# From Research to Practice: Ten Lessons in Delivering Digital Mental Health Services

**DOI:** 10.3390/jcm8081239

**Published:** 2019-08-17

**Authors:** Nickolai Titov, Heather D. Hadjistavropoulos, Olav Nielssen, David C. Mohr, Gerhard Andersson, Blake F. Dear

**Affiliations:** 1MindSpot Clinic and Department of Psychology, Macquarie University, Sydney, NSW 2109, Australia; 2Online Therapy Unit, Department of Psychology, University of Regina, Regina, Saskatchewan, SK S4S 0A2, Canada; 3Center for Behavioral Intervention Technologies, Department of Preventive Medicine, Northwestern University, Chicago, IL 60611, USA; 4Department of Behavioural Sciences and Learning, Linköping University, Linköping SE-581 83, Sweden; 5Department of Clinical Neuroscience, Karolinska Institute, Stockholm 171 77, Sweden

**Keywords:** delivery, implementation, internet-delivered cognitive behaviour therapy, psychological treatment, routine care, depression, anxiety disorders

## Abstract

There is a large body of research showing that psychological treatment can be effectively delivered via the internet, and Digital Mental Health Services (DMHS) are now delivering those interventions in routine care. However, not all attempts to translate these research outcomes into routine care have been successful. This paper draws on the experience of successful DMHS in Australia and Canada to describe ten lessons learned while establishing and delivering internet-delivered cognitive behavioural therapy (ICBT) and other mental health services as part of routine care. These lessons include learnings at four levels of analysis, including lessons learned working with (1) consumers, (2) therapists, (3) when operating DMHS, and (4) working within healthcare systems. Key themes include recognising that DMHS should provide not only treatment but also information and assessment services, that DMHS require robust systems for training and supervising therapists, that specialist skills are required to operate DMHS, and that the outcome data from DMHS can inform future mental health policy. We also confirm that operating such clinics is particularly challenging in the evolving funding, policy, and regulatory context, as well as increasing expectations from consumers about DMHS. Notwithstanding the difficulties of delivering DMHS, we conclude that the benefits of such services for the broader community significantly outweigh the challenges.

## 1. Introduction

Globally, mood and anxiety disorders affect more than 700 million people each year and are associated with considerable burden and disability [1,2]. However, in a 12-month period, fewer than half of those affected seek or receive evidence-based treatments [3,4,5] for reasons that include cost, limited availability of services in many areas, limited awareness of both illness and the potential benefit of treatment, stigma and preference to self-manage [6].

Delivering psychological services via the internet is one way of increasing access to care. A large number of randomised controlled trials have demonstrated that internet-delivered cognitive behavioural therapy (ICBT) is effective at treating anxiety and depression [7,8,9,10]. However, attempts to extend ICBT to routine care have produced mixed results. Several attempts at implementation have either been unsuccessful [11] or were not found to have added value to existing face to face services [12], which has raised doubt as to whether internet-delivered psychological services can be implemented successfully in typical health care settings [13]. Notwithstanding the challenges, the successful use of ICBT as part of routine clinical care has been reported in Sweden [14,15,16,17], the Netherlands [18,19], Norway, Denmark [20,21], Canada [22,23,24,25], and Australia [26,27,28,29,30]. In addition to reports of outcomes from individual clinics, and reflecting the maturing state of the field, there is now an increasing number of studies describing barriers [20,23,31,32], guidelines for implementation [33,34], and comparisons of clinics across different countries [35]. The successful clinics typically deliver ICBT interventions via so-called virtual or digital mental health services (DMHS). Common features of the successful clinics include high standards of both clinical and organisational governance, and robust systems for staff training and supervision [35].

This paper describes key lessons learned during our own efforts to develop and deliver DMHS. The MindSpot Clinic, Australia, and the Online Therapy Unit (OTU) in the province of Saskatchewan, Canada accept referrals directly from consumers as well as via general practitioners. Together, these DMHS have provided assessments to more than 100,000 people and treatment to more than 30,000 people. The authors have worked closely together for several years and have served on advisory bodies to each other’s services. The lessons we describe draw on our shared experiences in service development, delivery and collaboration. 

We narrowed down our experience to ten key lessons that were not fully described in other papers. These lessons were not immediately apparent to us when we set about translating our research findings to routine care but have been of fundamental importance in how we developed and now operate our DMHS. Hence, we anticipate that these lessons may help those launching similar clinics. 

We intentionally avoided specific frameworks of reporting [36,37], because an aim of this paper was to describe the experiences of operating mature services, rather than just the implementation phase. We also acknowledge that some of these lessons overlap, may not apply in other jurisdictions or even to other DMHS within our own countries.

## 2. Lessons

We chose to organise our lessons according to the model shown in Table 1, which represents the lessons learned from working with (1) consumers, (2) therapists, (3) operating DMHS, and (4) when operating in the broader health systems, including when engaging with funders and policy makers. 

Before further describing the lessons learned, key aspects of the MindSpot Clinic and OTU are summarised below. 

### 2.1. The MindSpot Clinic, Sydney, Australia

MindSpot was launched in 2013 and operates from Macquarie University, Sydney. MindSpot is funded by the Australian Government Department of Health, with funding initially provided for a 3-year period as a result of the competitive tender process. MindSpot aims to improve access to evidence-based education, triage, assessment, referral, and treatment services throughout Australia to adults with symptoms of depression and anxiety [28,29]. Clinic services are provided free of charge.

Patients can either self-refer after learning about MindSpot via the website (mindspot.org.au), online advertising, links from other mental health websites, recommendations by previous users or can take up referrals from health professionals. Patients first register online or via telephone and complete a detailed assessment questionnaire followed by telephone or secure email contact with a therapist to discuss symptoms and treatment options. Patients then choose between information to assist with self-management, referral to another service or ICBT. The clinic offers seven ICBT programs that have been validated in clinical trials, including transdiagnostic treatments designed to treat symptoms of anxiety and depression in several age groups [24,38,39,40,41,42,43,44,45,46,47] and disorder-specific treatments for obsessive compulsive disorder [48,49], post-traumatic stress disorder [50], and chronic pain [51]. All the treatment programs comprise of five lessons which provide the core information, delivered over eight-weeks. Additional resources targeting specific symptoms or difficulties are made available during treatment to assist patients tailor treatment to their own needs. Outcomes are measured using validated symptom scales that are administered weekly during treatment, on completion, and at a three-month follow-up. The therapists are all registered or provisionally registered mental health professionals who contact and monitor participants weekly during treatment via a secure email system or by telephone. The treatment patients are enrolled in cohorts every two weeks, with therapists each responsible for 50 or so patients. To date, more than 100,000 people have registered to use the clinic, and 25,000 have opted to receive ICBT.

### 2.2. Online Therapy Unit, Canada

The OTU has operated from the University of Regina in Saskatchewan since October 2010. Initial funding was provided by a federal research grant, but since 2015, the OTU has received stable funding from the Saskatchewan Ministry of Health. The OTU aims to provide therapist-guided ICBT for depression and anxiety and to educate providers of mental health care and conduct research on ICBT in routine practice [25]. Clinic services are also provided to patients free of charge.

The OTU promotes services to patients via word of mouth primarily from health care providers, media reports, and both digital and print communication. Patients are encouraged to visit the clinic online (onlinetherapyuser.ca) and can either self-refer or are referred by a health professional. Patients first complete an online screening followed by telephone assessment. 

The OTU delivers several ICBT programs including an adaptation of the Wellbeing course developed at Macquarie University and used at the MindSpot Clinic [44,47,52]. Clinically validated patient reported outcome measures (PROMS) of anxiety and depression are administered regularly during treatment, at post-treatment, and at three-month follow-up. Therapists are registered mental health professionals or graduate students under supervision employed by the clinic or by publicly funded community clinics located in other parts of Saskatchewan. During treatment, patients receive weekly therapist contact primarily via secure email or by telephone to assist in applying the skills taught during treatment. Since October 2010, the clinic had assessed more than 5400 patients, 4200 of whom have received ICBT [23,24,35].

## 3. The Ten Lessons

### 3.1. Level 1: Lessons Working with Consumers 

The first three lessons refer to the way DMHS can improve access to care and serve a broad section of the community, and how they deliver services other than treatment. 

#### 3.1.1. Lesson 1: DMHS Can Improve Access to Care for Those Who Really Need Care 

One of the most profound lessons we learned is that DMHS improve access to mental health services for many people who would not otherwise seek care. For example, at least a third of the users of both MindSpot and OTU report that they have not previously spoken to a health professional about symptoms and 80% of MindSpot and 55% of OTU users are not using other mental health services at the time of assessment [28,53]. 

The users of these services often have chronic and disabling symptoms. A third of MindSpot users have been troubled by their symptoms for between one and five years with a further third reporting symptoms for more than six years. Among users of the OTU, 45% report symptoms for two or more years and those in paid employment report an average of 7 days off work in the last 30 days due to symptoms. Moreover, the mean symptom scores reported by users of MindSpot are in the moderate–severe range, a quarter report suicidal thoughts and 2.4% disclose suicidal plans. 

We were also surprised at how people use our DMHS. While most users access content online, many others download information and refer to it when they do not have internet access [54]. In addition, for those who prefer or require printed materials, we send all the treatment course content in printed form [28]. For these reasons, we sometimes think of our services as delivering *virtual* rather than *digital* care, however, we are reluctant to introduce new terms to describe this already definitionally challenged field. 

#### 3.1.2. Lesson 2: DMHS Deliver More than Treatment 

Contrary to our expectations, not all consumers using our DMHS seek treatment, especially in Australia, where most users report they are primarily seeking a confidential assessment and recommendations about treatment options. This should not have come as a complete surprise, given the high proportion who had not previously sought treatment. However, we learned that for many people, the assessment itself serves as a brief but helpful clinical intervention, particularly when the consumer and therapist create a shared clinical formulation, discuss treatment options that include self-help strategies, and explore barriers and facilitators to recovery. This assessment process is highly regarded by the patients of both clinics. 

Consumers also frequently report enormous difficulty understanding and navigating the existing mental health service eco-system and some report using our services to seek advice about other health services, a theme which we return to below. Consequently, our second lesson is that DMHS need to offer a range of services in addition to online treatment, including information, assessment, triage and referral to other services. It should be noted that there are differences between our two clinics, with more people using the OTU reporting they are seeking treatment compared with users of MindSpot [55]. Thus, the specific services offered by DMHS may vary between jurisdictions, possibly reflecting differences in how the clinics are perceived and promoted, and differences in the needs of the local community. 

#### 3.1.3. Lesson 3: DMHS are used by a Broad Cross Section of the Population

Our third lesson is that a diverse cross section of our communities access our services, including indigenous Australians and Canadians [56], people on low incomes, people living in rural and remote regions, and other groups who often under-utilise traditional health services. We stress that our DMHS are not a panacea in this regard. Instead, we note that the widespread use of internet-enabled devices to access a range of services, including education, banking, entertainment and other domains, has extended to healthcare. In a similar way, DMHS have considerable potential to reduce, to some degree, the inequality of access to mental health. 

A striking example is the ability of DMHS to reach people living in rural areas. Almost 40% of MindSpot users report living outside major metropolitan areas, with many living in rural or remote parts of Australia, including islands off the mainland. Similarly, 28% of OTU users are from rural locations and 32% from small cities. Collectively, these people live in locations where access to health services is limited or sometimes non-existent [55]. A further example relates to engagement by older adults. Across both clinics, approximately 6% of users are over the age of 60, which is a group that often experience difficulty accessing mental health services, including for reasons related to physical impairments. The experience at both clinics is that older adults strongly engage in treatments and often obtain large improvements in symptoms [42]. 

A final example relates to socio-economic disadvantage. A recent analysis of MindSpot data found that users come from all socio-economic backgrounds, including 33% from the lower four deciles of socio-economic status, a group who are more prone to experience a disadvantage and difficulties accessing mental health services. 

### 3.2. Level 2: Lessons Working with Therapists

Both clinics have now trained large numbers of therapists, previously experienced in delivering face-to-face care, to deliver DMHS. Key lessons have been that therapists working in DMHS require specialised skills and therefore, require specialised training and supervision to acquire and maintain those skills.

#### 3.2.1. Lesson 4: DMHS Require Specialised Therapist Skills

An important lesson is that the skills and knowledge required for effective delivery of DMHS are sufficiently different to those associated with traditional models of care to warrant specialised training and supervision [57]. Obvious examples include the need for DMHS therapists to become competent in the use of clinical software platforms and in engaging with patients via telephone or text-based communication, including responding appropriately to very long messages or technical questions, skills that are not taught in most clinical training programs [23,40,58]. 

However, there are less obvious reasons for providing specialised training and supervision. First, therapists new to DMHS often intellectually understand that DMHS treatments can result in similar outcomes to face-to-face treatment [10,59] but may initially expect DMHS treatments to produce poorer outcomes, a sentiment they may communicate to consumers [23]. New therapists also often find that their assumptions about mechanisms or facilitators of recovery are challenged, particularly when they learn that DMHS patients can develop very strong therapeutic alliances [60] and may obtain large clinical improvements, even after choosing not to have regular therapist contact [55]. Related to this, therapists are often surprised to learn that the structured educational aspects and resources associated with DMHS offer significant advantages over traditional clinical care where such resources may not be used.

A second reason for providing specialised training and supervision is to support DMHS therapists to successfully process the transference and other dynamics that occur when delivering DMHS to large groups of patients at once. An example of this is the temporary increase in symptoms experienced by many consumers at mid-treatment when they begin to apply skills learned in treatment in their everyday lives. These lapses usually resolve and most continue to recover. This process is familiar to experienced therapists, who can help patients understand this trajectory and can manage their own reactions. However, the effect can be magnified by the large numbers of patients in each treatment cohort, leading to feelings of intense elation or sometimes disillusionment, particularly when some patients choose not to engage with the therapist but have not made this clear at the outset. A strong framework of training and supervision can assist therapists to understand and adapt to these patterns and maintain confidence in both their own performance and the effectiveness of the treatment programs. 

#### 3.2.2. Lesson 5: DMHS Require Specialised Clinical Processes

This lesson reflects important differences in the clinical procedures and processes used in DMHS compared to traditional mental health clinics. An obvious example is the use of structured ICBT interventions, questionnaires and outcome measures used in DMHS, the delivery of which are governed by procedures which regulate therapists’ actions more than they would in typical in face-to-face services. 

This level of structure reflects how DMHS attempt to manage both quality assurance and treatment to large numbers of consumers. Therapists experienced in working at DMHS can efficiently deliver individualised care within these structured frameworks, but this is more difficult for less experienced therapists. Another lesson has been the importance of robust systems for not only training and supervision [58,61], but also the recruitment and retention of clinicians who are comfortable with relatively high levels of structure and process [23]. 

Another example of how DMHS differ from traditional face-to-face mental health services, at least in our jurisdictions, relates to the use of PROMS and patient-reported experience measures (PREMS) [62,63,64]. Despite the documented utility of PROMS and PREMS in clinical care, they are infrequently used in traditional services, and rarely as a therapeutic tool for guiding discussions or decisions about treatment or as a method for improving the quality of care. Since our clinics routinely administer PROMS and PREMS during and after treatment, we provide specific training for new therapists to increase their comfort and competence in using measures of outcome and experience [61]. 

### 3.3. Level 3: Lessons Operating Services

#### 3.3.1. Lesson 6: The Operation of DMHS Require Specialised Systems and Skills 

This lesson is obvious, but we include it here because we underestimated the complexity of developing and delivering safe and effective DMHS. We expected that our DMHS would be similar to traditional face-to-face clinical mental health services or an extension of the operations used in our large-scale clinical research trials. However, we quickly learned that safely and effectively operating DMHS required attention in at least four areas.

First, DMHS require robust procedures to define and effectively manage safety risks for people presenting with more severe and complex needs than seen in our clinical trials [30] and who often live in remote locations. This requires developing expertise in evaluation of risk via telephone or online communication, the ability to contact emergency services that are available near where the patient is located, how to refer to such services, and how to stay abreast of changes in their referral and contact details. 

Second, although operating DMHS involve similar skills as those required for operating traditional mental health services, including management, human resources, marketing and IT [25], we found that DMHS were sufficiently different to warrant employing people with additional expertise, including skills relevant to telehealth, social media and online marketing.

Third, and in addition to the urgent requirement for establishing robust systems of organizational and clinical governance we recognised that in order to effectively lead our services, we personally needed to develop commercial and management skills, domains in which we, as clinical researchers, had little or no experience. We also needed to address challenges relating to regulation; for example at MindSpot, we needed to determine which of the myriad of possible regulatory frameworks applied to our activities [65] given that most [66] had been developed for traditional face-to-face services. Within the OTU, a similar issue requiring attention was how to meet the different regulatory requirements for services provided by therapists from psychology and social work. 

Establishing, maintaining, and subsequently improving our operational systems has required a considerable work effort. Whilst we have recruited specialist staff to assist with such efforts, due to the novelty of DMHS, we often also sometimes found it necessary to train and develop our own staff in these operational and managerial roles.

#### 3.3.2. Lesson 7: Digital Mental Health Clinics Evolve 

This lesson represents another difference between DMHS and traditional clinical services. Traditional services may change frequently, but the change is usually limited to organisational structure or branding, with less frequent changes to the service or delivery models. By contrast, our DMHS regularly undergo significant changes in procedures, systems, and even service delivery models due to developments in research, technology, the changing expectations of consumers, and changes in the policy priorities of the funding bodies. However, the most frequent changes stem from reviewing our outcomes and procedures.

For example, within the OTU, changes within recent years include (1) replacing disorder-specific ICBT programs with a transdiagnostic treatment program in light of the extensive comorbidity found among patients seeking services and the efficiency of delivery compared to disorder-specific programs [24], (2) ongoing trials to determine the best level of support and specialisation by therapists [40,67], and (3) the expansion of services to address other needs in the community, such as ICBT for pain [22]. 

Regular change has significant implications for the operation of our clinics. For example, management needs systems for reliably collecting and analysing data, the ability to develop and test alternative models, and the skills and authority to make decisions. The individuals given the task of implementing change need project leadership and change management skills and procedures to plan and deploy changes. Furthermore, therapists and other staff need to be prepared for and willing to implement changes. This means that all staff members need to be adaptable and change needs to become part of the culture of the DMHS. 

### 3.4. Level 4: Lessons Working with Health Systems, Funders, and Policy Makers

This final group of lessons summarises key learnings derived from working with and influencing health systems, in particular, the future role and value of DMHS within health systems. 

#### 3.4.1. Lesson 8: Integrating DMHS within Health Systems Is Challenging

Our DMHS reside within enormously complex health systems which might be more accurately described as interconnected nodes of care rather than true *systems*. For example, mental health services, while all purporting to share the aim of improving mental wellbeing, often target different groups and may be accountable to different policy and regulatory frameworks. Health services also differ with respect to funding, and in both Canada and Australia, funding for different types of mental health services can be provided by the federal government, state/provincial government, state mental health commissions, non-governmental organisations and individual consumers. As a result, mental health services are often fragmented, poorly connected, and difficult for consumers to navigate. 

The complexity of health systems and their resistance to change created a number of threats to the sustainability and stability of our DMHS, particularly in our early years of service delivery. We were able to overcome such challenges by building strong relationships with key stakeholders and in particular, by publishing outcome data that documented the value of the services. This data has also assisted in defining the role of DMHS in the broader mental healthcare system, including as services which improve access to patients who would otherwise not access mental health services.

These so-called external-facing activities have required considerable time and effort. Participation in such activities has required us to learn the sometimes-subtle rules of engaging with other organisations, to commit to regular participation in networking activities, and to make frequent efforts to build and maintain collaborations. Such activities can be enormously time consuming but we have found that they are an essential component of successfully delivering our DMHS. 

#### 3.4.2. Lesson 9: DMHS May Change the Mental Health System

By providing services to large numbers of consumers and because of our routine collection of outcome data, our clinics are having a growing influence on the health system and are increasingly seen as agents or examples of change. In Canada, the experiences and activities of the OTU have influenced the development of e-mental health in several ways [68]. In Australia, data from MindSpot has helped inform long-term government planning and funding strategies for mental health services [69], not only DMHS. The data and outcome driven reports prepared by our services are often in marked contrast to submissions by other groups, which may be based more on opinion rather than on evidence. Hence, an important role of DMHS is to inform policy makers and funders to help improve the broader mental health system by presenting data drawn from a broad cross section of the community and also to nudge traditional services to adopt systems of measurement and reporting of outcomes. 

#### 3.4.3. Lesson 10: DMHS Are Not a Panacea

We are struck by how often mental health funders and policy makers, when presented with the evidence from our DMHS, become enthusiastic about their potential without an appreciation of their limits. These limits include the so-called digital divide, that is, the group in society who does not use digital devices, those with very low levels of literacy, and those in crisis, who can benefit from contact with DMHS, but may be better off with a mental health service that includes direct human interaction, as well as those who prefer to see someone face-to-face. 

In all our communication, we emphasise that DMHS should complement and not replace existing services. We also emphasise that attention must be paid to systematically evaluating delivery methods that combine the best elements of DMHS with traditional services, including blended care [70,71,72,73,74]. We also stress that consumer knowledge of DMHS is still limited and that even brief education about DMHS can improve consumer perceptions and uptake of services [75]. These observations lead to our final lesson that whilst acknowledging that DMHS can significantly improve access to safe, clinical and cost-effective care, our DMHS are not a panacea.

## 4. General Discussion

This paper aimed to assist other emerging DMHS by sharing ten lessons we learned from successfully delivering DMHS to very large numbers of consumers. Some of these lessons might seem obvious, but their importance was not always apparent when we started our services. Several key themes are discussed below, followed by recommendations.

One theme is that we expect that demand for this service model will grow. The number of patients treated using ICBT in the OTU has more than doubled in the past four years. The threshold for accessing this model of care is significantly lower than traditional face-to-face services and consumers are becoming increasingly comfortable with using technology to access a broad range of services, including health services. Along with a growth in numbers, we expect that existing DMHS will become more tailored to different populations, for example, people in certain occupations, different cultures, or who have been referred from different pathways, although our experience is that the extent to which the treatment course materials need to be customised is considerably less than expected [56,76,77,78].

A second theme is that the workforce requires specialised training, clinical supervision and support. This raises broader issues about workforce planning and training programs. We note that many professional bodies have recognised the importance of education and training of mental health professionals and standards in this model of care [79,80,81,82] but that few training programs in any mental health discipline offer courses or training opportunities specifically for digital mental health. The absence of such training opportunities poses significant risks for the future sustainability and quality of the field. 

A third theme is that the delivery of DMHS requires specialist skills in both clinical and operational domains. We also note that although the costs to entry of developing a DMHS, especially a low volume service, might be relatively low, the costs of maintaining quality services can be high. Inadequate funding and inadequate organisational governance can affect the reputation, credibility, and therefore, the potential of the emerging field of DMHS [83]. Hence, we strongly encourage anyone seeking to launch a DMHS to carefully consider the governance frameworks that will ensure the safe and sustainable delivery of services, or to consider licensing their interventions to groups who have proven success in implementing similar services. 

Another theme is that the field of DMHS is rapidly evolving. We encourage those seeking to start a DMHS to consider trialing different models of care to those currently used by existing DMHS, including testing different levels of therapist support [24,40,55] and testing care which combines both face-to-face and online delivery. We note the important work conducted by our European colleagues on blended care [70,71,72,73,74] and by others on mobile services [13] and encourage collaboration in order to collectively develop the most effective models of care. 

Our final theme relates to recognising the true value proposition of DMHS. We maintain that they are not a panacea but instead serve several valuable functions, including as a useful complement to existing services, as a way of improving equity of access to mental health care for common psychological disorders, and as a stepping stone to other services. Over-promising may increase the likelihood of short-term funding, but poorly designed and delivered services might harm consumers, disappoint stakeholders and risk the future of DMHS. 

### 4.1. Recommendations

These observations lead to several recommendations which we encourage those contemplating developing DMHS to consider. First, we recommend that new DMHS recruit not only appropriately skilled therapists, but also people with commercial and professional skills, ideally with experience in digital service delivery. Second, given the unique challenges of DMHS, we recommend the development of both thorough initial training of therapists, as well as of systems for ongoing training and supervision. Third, given the likelihood that demand for DMHS will grow, we strongly encourage that organisations involved in training and certification of mental health professionals add content and training opportunities relevant to the competencies required in DMHS. 

Fourth, we recommend that emerging DMHS measure and publish their outcomes, including disappointing and negative effects outcomes [84]. We also encourage DMHS to engage with policy makers and funders to develop mental health policy grounded in evidence rather than in opinion. Finally, we strongly encourage DMHS to engage with their consumers in appropriate co-design and evaluation activities to ensure services are not only effective but acceptable to consumers. 

### 4.2. Strengths and Limitations

We believe that the main strength of this paper stems from the authors’ shared experience in launching and steadily improving successful high volume DMHS. However, we acknowledge several weaknesses, including that the list of lessons is non-exhaustive and did not include some of the significant challenges associated with managing funding insecurity or bureaucratic and professional challenges within the field of mental health, topics we will return to in a subsequent publication. We also acknowledge that our experiences may not reflect those of other DMHS. 

### 4.3. Conclusions

This paper described ten key lessons learned by the authors when developing, delivering, and evaluating DMHS. Despite the challenges, provided they are delivered safely, effectively and with strong clinical, operational and organisational governance, we remain highly optimistic about the potential of DMHS to reduce the global burden of the high prevalence of mental disorders. 

## Figures and Tables

**Table 1 jcm-08-01239-t001:** Lessons learned at four levels of analysis of digital mental health services (DMHS).

Level	Lesson
1. Consumers	DMHS can improve access to care for those who really need care DMHS deliver more than treatment services DMHS are used by a broad cross-section of the population
2. Therapists	DMHS require specialised therapist skills DMHS require specialised clinical processes
3. Operating DMHS	The operation of DMHS require specialised systems and skills DMHS evolve
4. Health Systems, Funders, and Policy Makers	Integrating DMHS within health systems is challenging DMHS may change the mental health system DMHS are not a panacea

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
