# Peer review of "From Research to Practice: Ten Lessons in Delivering Digital Mental Health Services"

_jcm, 2019, doi:10.3390/jcm8081239_

Round 1

Reviewer 1 Report

This manuscript is a very interesting contribution to the community engaged in providing digital tools for Mental Health. While, as the authors state, some of the lessons presented might seem obvious, it is important to openly provide them to the community, particularly because these are grounded on the authors' experience in creating, delivering and assessing such services.

One of the main issues often associated to the assessment of the impact of these services is the lack of a objective report of the outcomes, whatever they are, to improve our knowledge of what works and doesn't. This manuscript is, in my humble opinion, a positive step, in this regard.

I would only suggest that the document is carefully proofread, to eliminate a few issues.

Author Response

We thank the reviewer for their positive comments and helpful recommendation. We have now carefully proofread the manuscript and have submitted the edited version, with track changes. 

Reviewer 2 Report

Thank you for the opportunity to be among the first to read this very well written manuscript and sharing your experience. It will be of great help for anyone engaging or wanting to engage in the provision of DMHS.

It is relevant and interesting. To my knowledge, this is the first manuscript summarizing lessons learned in delivering digital mental health services in multiple health systems, to multiple populations. As the authors point out, some of the points have been made separately before, but the summary is extremely helpful and relevant to the field and also to policy makers. The paper is well written and the text is clear and easy to read.

I have only two very minor comments:

line 318: You may want to check that sentence

references with organisational authors need to be checked and corrected

Author Response

We thank the reviewer for the positive comments and helpful recommendations.

Regarding:

1. line 318: You may want to check that sentence. This sentence has now been edited

2. References with organisational authors need to be checked and corrected. Thank you for alerting us to these, which have now been checked and corrected

Reviewer 3 Report

This paper described ten key lessons learned by the authors when developing, delivering, and evaluating DMHS. It is a good paper with an excellent revision of the bibliography on the subject. One of the main strength is that the authors’ shared experience in launching and steadily improving successful high volume DMHS.

This paper aims to assist other emerging DMHS by sharing nine-ten lessons authors have learned from successfully delivering DMHS to very large numbers of consumers. Also, Authors included recommendation list that helps all people to use DMHS.

I recommend it´s publication as it is with these two little minor changes:

-4 levels are adequate and well justified, although some of the lessons seem redundant and can be reduced to fewer lessons (in the 4th level only 2 lessons for example)

-The table should reflect the lessons with the number in which they appear in the text (1 to 9 or 10).

Author Response

We thank the reviewer for their kind comments and helpful recommendations.

Re:

-4 levels are adequate and well justified, although some of the lessons seem redundant and can be reduced to fewer lessons (in the 4th level only 2 lessons for example)

We appreciate the reviewer's comments and have revised one of the three lessons in the 4th level to better emphasise its importance.  

-The table should reflect the lessons with the number in which they appear in the text (1 to 9 or 10).

We have now addressed this error.